# Investigating the Interplay between Cardiovascular and Neurodegenerative Disease

**DOI:** 10.3390/biology13100764

**Published:** 2024-09-26

**Authors:** Jason Patrick Cousineau, Aimee Maria Dawe, Melanie Alpaugh

**Affiliations:** Department of Molecular and Cellular Biology, University of Guelph, Guelph, ON N1G 2W1, Canada; cousinej@uoguelph.ca (J.P.C.); adawe03@uoguelph.ca (A.M.D.)

**Keywords:** human studies, animal models, combinatorial models, renin–angiotensin system, autonomic nervous system, cerebral blood flow, blood–brain barrier, behaviour

## Abstract

**Simple Summary:**

Traditionally, models of neurodegenerative diseases focus on changes within the central nervous system. However, growing evidence indicates there are prominent interactions between the cardiovascular system and neurodegenerative diseases. In this review, we discuss cardiovascular impairments in models of neurodegeneration, neurological impairments in models of cardiovascular disease, and what occurs when the two are combined as a first step toward improving our understanding of the pathophysiology of neurodegenerative diseases.

**Abstract:**

Neurological diseases, including neurodegenerative diseases (NDDs), are the primary cause of disability worldwide and the second leading cause of death. The chronic nature of these conditions and the lack of disease-modifying therapies highlight the urgent need for developing effective therapies. To accomplish this, effective models of NDDs are required to increase our understanding of underlying pathophysiology and for evaluating treatment efficacy. Traditionally, models of NDDs have focused on the central nervous system (CNS). However, evidence points to a relationship between systemic factors and the development of NDDs. Cardiovascular disease and related risk factors have been shown to modify the cerebral vasculature and the risk of developing Alzheimer’s disease. These findings, combined with reports of changes to vascular density and blood–brain barrier integrity in other NDDs, such as Huntington’s disease and Parkinson’s disease, suggest that cardiovascular health may be predictive of brain function. To evaluate this, we explore evidence for disruptions to the circulatory system in murine models of NDDs, evidence of disruptions to the CNS in cardiovascular disease models and summarize models combining cardiovascular disruption with models of NDDs. In this study, we aim to increase our understanding of cardiovascular disease and neurodegeneration interactions across multiple disease states and evaluate the utility of combining model systems.

## 1. Introduction

According to recent global burden of disease studies, neurological disorders affect approximately 43.1% of the global population, are the second leading cause of death, and are the leading cause of disability [1,2]. As the population ages, an increasing proportion of these incidences are classified as neurodegenerative diseases (NDDs). NDDs present a major societal challenge due to their extended duration and life-altering impact, resulting in a financial burden amounting to billions of dollars for healthcare systems [3,4]. 

To counteract this problem, it is necessary to improve our understanding of disease processes such that appropriate animal models can be developed and used to identify disease-modifying therapies or preventative strategies. One promising area of research is the current focus on understanding common risk factors that impact multiple disorders such as cardiovascular disease (CVD) [5,6,7,8,9,10]. Interestingly, comorbid CVD is often highlighted as a major risk factor for NDDs and the two appear to be highly intertwined [11,12]. Therefore, this review will summarize the current understanding of the relationship between CVD, CVD risk, and several NDDs. It concludes with a discussion on how existing pre-clinical models of these two groups of diseases may be combined to improve our understanding of their interactions. 

## 2. Human Studies

### 2.1. Neurodegenerative Diseases

Substantial evidence exists in the literature regarding interactions between cardiovascular diseases and NDDs. This includes both an increased probability of developing NDD in those with pre-existing cardiovascular disease, as well as an increased risk of developing certain CVDs as a part of the pathophysiology of multiple disorders [13,14,15,16,17,18,19,20,21,22]. In this section, we will discuss the associations between CVD and Alzheimer’s Disease, Parkinson’s Disease, Multiple Sclerosis, Amyotrophic Lateral Sclerosis, and Huntington’s Disease.

#### 2.1.1. Alzheimer’s Disease

Alzheimer’s disease (AD) is the most prevalent NDD and the leading cause of dementia globally [23]. Key disease features include cognitive decline, accumulation of amyloid-beta and tau proteins, and neuronal death across various brain structures, with the hippocampus being the most highly affected [24,25]. These hallmarks of pathophysiology are also coupled with multiple changes to the cerebral vasculature and cardiovascular regulation, such as decreased cerebral blood flow, abnormal vasculature, impairments to the blood–brain barrier, and an attenuated renin–angiotensin system (RAS) [13,14,26,27]. The current prevailing theory is that these pathophysiological processes begin more than 20 years before disease onset [3,24,28]. This theory aligns well with longitudinal studies showing that changes to blood pressure in midlife increase the risk of AD. For example, hypertension in midlife has been found to increase the risk of developing AD by 20% per 10 mm Hg increment increase in systolic blood pressure, and hypertension sustained from midlife into late life is associated with as much as a 25% increase in the risk of developing AD [12,29]. In addition to influencing onset, hypertension during early stages of disease exacerbates the progression of AD, leading to a greater negative effect on cognition [30,31]. As disease pathology progresses, the relationship with blood pressure becomes more complex [32]. Still, most evidence points to a substantial enhancement of pathology during what are likely the pre-manifest periods of disease [33]. 

Interestingly, abnormal blood pressure is not the only systemic factor that has been linked to AD. Heart failure (HF), atherosclerosis (AS), and atrial fibrillation (AF) in mid-to-late life have all been documented to increase the likelihood of developing dementia in general and AD in particular [34,35,36,37]. In HF, blood-based biomarkers associated with AD, such as beta-amyloid, tau, glial fibrillary acidic protein (GFAP), and neurofilament-light can foreshadow cognitive decline and correlate to the severity of HF [38,39]. Additionally, EEG scans of AD patients reflect those of HF patients, with both showing indications of acute hypoxia which suggests a disruption in cerebral blood flow [40]. In AS, AD severity is also thought to increase with the severity of atherosclerotic plaques as they correlate with the density of cardinal neuropathological lesions seen in AD, such as phosphorylated tau spindles and amyloid-beta plaques, especially when the patient possesses the ε4 allele of the Apolipoprotein E gene (ApoE4) [41,42]. This relationship can also be seen within postmortem tissue, as atherosclerotic plaques in the circle of Willis have been found to have more extensive occlusions in AD patients as compared to those without AD [43]. In a similar fashion, it is common for those with AD to have some degree of cerebral amyloid angiopathy (CAA) due to disrupted lipid and beta-amyloid transport at the blood–brain barrier, in which it was found that AS may exacerbate amyloid deposits within the cerebral arteries and in turn, this can lead to increased cognitive deficits [44,45]. Within AF patients, AD was found to be more likely to develop in patients <70 years of age as compared to those >70 years of age and result in earlier cognitive decline across all patients in the study [46]. This suggests that AF may exacerbate AD development in younger individuals. 

While CVD is generally thought to impact AD in a unidirectional manner, there is some evidence that the relationship may be more complex. This is highlighted by the presence of shared genetic risk factors for CVD and AD, as well as evidence of AD-associated protein accumulation appearing in the hearts of AD patients. For example, mutations in the presenilin genes *PSEN1* and *PSEN2* associated with early familial AD are also associated with dilated cardiomyopathy (CM) and HF [47], while the ApoE4 has also been associated with ischemic heart disease, myocardial infarction (MI), and AS [11,48]. Furthermore, amyloid-beta aggregates have been identified within cardiomyocytes, disrupting Ca^2+^ homeostasis and contributing to myocardial dysfunction [49,50]. While it is not yet known how these protein aggregates arise, it is possible that they spread through the autonomic nervous system, as similar mechanisms have been previously reported in other conditions [51]. Alternatively, the conditions that drive protein misfolding in the brain may also apply to the heart and the aggregates may reflect a shared susceptibility to genetic risk factors, as mentioned earlier, or general risk factors such as aging. 

#### 2.1.2. Parkinson’s Disease

Parkinson’s Disease (PD) is the second most common NDD behind AD [52] and is characterized by Lewy body inclusions, death of dopaminergic neurons within the midbrain, abnormal vascular morphology, reduced cerebral blood flow, impairments to the blood–brain barrier and changes to the RAS [18,53,54,55,56,57]. Together, these pathological changes ultimately result in the development of multiple behavioural alterations including significant motor deficits in mid-to-late life [52,57]. Despite being largely seen as a motor disorder, PD is a multisystem disease with well-described disruptions to the autonomic nervous system, which frequently appear long before the onset of symptoms within the CNS. 

Common examples of autonomic dysfunction in PD are orthostatic hypotension [58,59], which can be present in up to 58% of patients, supine hypertension [60,61,62,63], and reduced heart rate variability [64]. All three aspects of autonomic dysfunction have been linked to a general worsening of disease severity [64,65,66], as well as to a heightened risk of the development of overt CVD such as MI, stroke, left ventricular hypertrophy, HF, various arrhythmias, and CM [60,61,62,63,67,68,69]. As a result of these interactions, those with PD seem to be at an increased risk of MI and demonstrate higher mortality rates resulting from MI than controls [70]. These blood pressure and electrical abnormalities are thought to result from sympathetic denervation of the heart [71,72]. However, a significant number of genes associated with either mitochondrial or heart contractile function have been found to either be upregulated or downregulated in PD compared to non-PD controls, suggesting that there may be some degree of shared susceptibility to disease processes between heart disease and PD [73]. The presence of mitochondrial dysfunction in PD particularly supports this hypothesis as genetic disorders associated with mitochondrial disruption are commonly observed to have both cardiac and neurological impairments [74].

#### 2.1.3. Multiple Sclerosis

Multiple sclerosis (MS) is a chronic inflammatory NDD that results in the demyelination of the CNS [75] by pathogenic lymphocytes, T cells, and B cells that enter the CNS and induce local inflammation [76]. This strong inflammatory phenotype is unsurprisingly associated with alterations to the cerebral vasculature and cardiovascular regulation, including prominent disruptions of the blood–brain barrier, modified vascular morphology, and altered expression of RAS components in immune cells [77,78,79].

Similar to the previously mentioned NDDs, several relationships have been documented between MS and CVD. Epidemiological studies have highlighted an increased risk of MS patients developing CVD such as MI, stroke, HF, and AS [80,81,82,83,84], as well as a more rapid disability progression in those with either recently developed or pre-existing CVD [80]. This relationship can also be seen from a genetic standpoint, as a genome-wide association study involving almost 116,000 MS patients found significant genetic evidence for shared risk factors between MS and MI, HF, stroke, and AS disorders [85]. Overall, these associations support the idea of some degree of shared pathobiology between these disorders. These findings were further supported by a second genome-wide association study that found 133 pleiotropic loci alongside 60 standalone genes associated with both MS and CVD risk factors including gene loci associated with changes to low-density lipoprotein, high-density lipoprotein, triglyceride metabolism, and C-reactive protein amongst others [86]. Interestingly, AF seems to have no significant association with MS [84,85]. 

Aside from associations with various CVDs, MS has also been linked to autonomic dysfunction. For example, MS patients appear to have an impaired baroreflex, resulting in decreased heart rate variability and impaired vasoconstriction [87,88,89,90]. Since the baroreflex acts as a negative feedback loop, helping to maintain consistent blood pressure, this dysfunction offers an explanation for the increased prevalence of hypertension within the MS population as compared to the general population [91,92,93]. Not only is hypertension more prevalent in this patient population, but it, as well as HF, have also been reported to exacerbate disease progression, specifically regarding increased brain atrophy, cognitive impairment, and physical disability [80,94,95,96,97,98,99]. 

#### 2.1.4. Amyotrophic Lateral Sclerosis

Amyotrophic lateral sclerosis (ALS), is a rapidly progressing and fatal NDD that affects both upper and lower motor neurons, resulting in muscle weakness and wasting, ultimately leading to total paralysis [100]. Similar to the other discussed NDDs, the pathophysiology of ALS includes changes to the cerebrovasculature, such as impairments of the blood–brain barrier and the neural vascular unit, as well as dysregulation of cerebral blood flow [101,102]. Despite the overlapping changes to the cerebrovasculature and the similarity of risk factors between ALS and other NDDs, the relationship between cardiovascular disease and onset or severity of ALS is less clear with many published studies reporting contradictory findings. 

For example, a few studies have found evidence to support an association between ALS onset and hypercholesterolemia, an important risk factor for AS, with hypercholesterolemia reported to decrease the risk of developing ALS [103]. However, two prior studies found no association with AS, and that good cardiovascular health was hypothesized as being more associated with ALS rather than poor cardiovascular health [103,104]. Similarly, the findings surrounding hypertension are highly variable with significant regional differences being present [105]. To date, there is no clear consensus regarding how premorbid cardiovascular disease impacts the risk of developing ALS.

In contrast to this, the data surrounding comorbid cardiovascular disease are slightly clearer with multiple studies, including a meta-analysis, reporting that HF and AF tend to exacerbate disease and lead to reduced survival [105,106,107]. Currently, it is thought that this greater disease severity is linked to autonomic dysfunction which occurs as part of ALS pathophysiology. Specifically, patients have been reported to display an elevated resting heart rate and lower overall heart rate variability, as well as decreased cross-sectional area of the vagus nerve [108,109,110]. These changes are thought to result from a progressive denervation of the autonomic nervous system [111] or from the atrophy of regions of the hypothalamus involved in regulating the sympathetic nervous system. Given that heart failure and AF cause greater sympathetic activation, these comorbidities would exacerbate the underlying centrally mediated autonomic dysfunction in ALS and lead to an increased risk of severe cardiovascular failure and death. Evidence for this potential impact of ALS-mediated autonomic dysfunction on the cardiovascular systems comes from multiple case study reports of Takotsubo syndrome in ALS patients [112].

#### 2.1.5. Huntington’s Disease

Huntington’s disease (HD) is a rare, dominantly inherited NDD resulting from a trinucleotide repeat expansion in the huntingtin (*HTT*) gene, presenting in midlife with a triad of psychological, physical, and cognitive impairments [113,114]. Mutant huntingtin (mHTT) is ubiquitously expressed throughout the body and is more prone to misfolding and forming aggregates than the wild-type protein [115,116]. Despite being primarily studied as a disease of the CNS, the ubiquitous expression of mHTT has been described to result in numerous peripheral abnormalities in HD including pathological changes to the cardiovascular system, such as altered heart rate variability [117,118,119], increased resting heart rate, and the presence of protein aggregates within the heart [120,121]. While some of these changes seem to be mediated through central disruption of nuclei responsible for the regulation of the autonomic nervous system [122], the presence of cardiac fibrosis and aggregates within cardiac tissue points to a potential contribution of cell-autonomous dysfunction of cardiac tissue to cardiovascular impairments in HD [123]. 

In addition to these functional changes to the cardiovascular system, risk factors for heart disease such as hypertension [120,124] and hypercholesterolemia [125] are also modified in the HD population. In the case of hypertension, there is some controversy in manifest patients with some studies reporting an increase [124], while others report a decrease [126]. However, an overall increase in systolic blood pressure is apparent in premanifest patients as early as 25 years prior to clinical onset of disease [120]. Furthermore, there is some evidence that the RAS is modified in the brains of HD patients [127,128]. In contrast, the rates of hypercholesterolemia are decreased, with most studies reporting lower levels of cholesterol and AS-associated lipoparticles compared to non-gene carriers [125,129]. The effect of these changes on overall health or disease progression is not yet fully elucidated. However, there is evidence that decreased heart rate variability increases the risk of falling [130], and that hypertension may modify the age of onset and/or symptom severity [126,131,132,133,134]. 

These peripheral changes to the cardiovascular system are accompanied by disruptions in the cerebral vasculature, which include reduced cerebral blood flow [16,135,136,137], morphological abnormalities [138], and blood–brain barrier deficits [16]. It is not yet known how these changes to the cerebral vasculature impact disease; nonetheless, imaging studies have shown that reduced cerebral blood flow is associated with more severe cognitive and motor symptoms [15,137,139,140,141], suggesting that these changes relate to greater disease severity. 

#### 2.1.6. Summary

All the discussed NDDs have clinical evidence of interactions with CVD, with changes to the cerebral vasculature, including blood–brain barrier impairments and altered cerebral blood flow, being nearly universal features of these conditions. Additionally, almost all neurological disorders have been linked to autonomic dysfunction which may exacerbate any underlying cardiovascular disruptions. Beyond the presence of changes to the vascular system of the brain and central regulation of the heart, CVD and corresponding risk factors have complex interactions with NDDs. In the case of AD, HD, and ALS, there is evidence that CVD either initiates or influences disease severity. In contrast, MS and PD seem to share risk factors with cardiovascular disease that suggests their underlying pathobiology may lead to disruption of both the brain and the heart. These shared features appear to influence the prevalence of cardiovascular dysfunction which may then in turn exacerbate disease. Regardless of the direction, clinical studies provide strong evidence of interrelations between these classes of disorders. To further explore this possibility, the evidence of changes to the CNS after primary CVD diagnosis is discussed in the next section.

### 2.2. Cardiovascular Diseases

CVD has been well documented to have detrimental effects on the entirety of the human body, including on the healthy brain. The specific effects vary based on the type of CVD, therefore, the common cardiovascular risk factors hypertension, AS, as well as the common forms of CVD; AF, MI, CM, and HF will each be discussed in the following section. 

#### 2.2.1. Hypertension and Atherosclerosis 

Hypertension is one of the most common disruptions to the cardiovascular system (high blood pressure, >140 mm Hg) and although it is not in itself considered a CVD it is an important risk factor for the development of various CVDs which have numerous effects on the body [142]. Importantly, hypertension has been described to have a number of detrimental effects on the vasculature of the central nervous system, including increased blood–brain barrier permeability [143], altered vessel diameter, increased lumen-to-wall ratio [144], and decreased cerebral blood flow [145,146]. Of these changes, reduced cerebral flow has been particularly associated with cognitive decline [145,146,147,148,149], which is thought to result from a hypo-perfusion-induced thinning of the cerebral cortex [150,151,152,153].

Much like hypertension, AS, is the leading pathological process involved in the development of most CVDs and is defined as the thickening of arterial walls by 1 mm or more, with the accumulation of plaque deposits that encroach into the lumen [154,155]. AS can influence the brain by reducing cerebral blood flow and cerebrovascular reactivity within multiple regions, including both the subcortical structures of the basal ganglia and the superior frontal lobe [156]. Additionally, imaging analyses of the total brain volume of AS patients have shown a decrease in the grey matter volume of the cortex as compared to age-matched controls [157]. 

#### 2.2.2. Myocardial Infarction

MI is a form of CVD that results in a complete or partial blockage of blood from accessing the myocardial tissue of the heart, resulting in irreversible necrosis [158,159]. This necrosis reduces the heart’s ability to properly contract and perfuse the body with the necessary blood supply [160,161], which can lead to hypoperfusion in survivors of MI despite their blood pressure being within the ideal range [162]. When examining the effect of MI on cognition, studies have found that there are mild decreases in the cognitive ability of individuals post-MI [163]. While the mechanisms of this interaction are not fully understood, a current theory suggests that cognitive decline may in part be explained by systemic inflammatory responses [164,165]. 

#### 2.2.3. Atrial Fibrillation

AF is the most common type of arrhythmia which is characterized by a quivering of the atrial chamber of the heart, resulting in an irregular heartbeat [166,167]. A major point of concern with AF is the increased risk and severity of stroke [168,169,170] as well as decreased cerebral blood flow [171]. Additionally, a smaller total brain volume, increased quantities of white matter lesions and cortical infarcts, memory impairments, reduced processing speed, and diminished executive function have all been associated with AF in stroke-free cohorts [172,173,174,175]. 

#### 2.2.4. Cardiomyopathies

CMs are a group of disorders that cause functional and structural abnormalities of the heart. Within the literature, CM may be referred to as HF but can be divided into several distinct subcategories including dilated CM, hypertrophic CM, restrictive CM, arrhythmogenic CM, and takotsubo CM, with varying prevalence depending on the sub-type [176,177,178,179,180,181,182]. Collectively, all varieties of CM can influence the vasculature of the body, resulting in ventricle remodelling, decreased blood pressure, decreased peripheral blood flow, and altered cerebral blood flow [183,184,185,186,187]. Due to the vascular changes, CM can increase the risk of sudden cardiac death, HF, and AF [184,185,188]. Additionally, cognitive impairments such as dementia, and cerebral atrophy have also been attributed to CM [189,190].

#### 2.2.5. Heart Failure

HF is another form of CVD that causes a significant challenge to the healthcare system. HF is currently one of the fastest-growing CVDs globally, with the total number of cases increasing by 12% from 2002–2014 [191,192]. There are three varieties of HF which are differentiated by the volume of blood ejected by the left ventricle after each contraction. HF with a preserved ejection fraction (HFpEF) results when the heart retains an ejection fraction (EF) of 50% or greater, HF with a mildly reduced ejection fraction (HFmrEF) results when the heart retains only 40–49% of its EF and HF with a reduced ejection fraction (HFrEF) results when the heart retains less than 40% of its EF [193]. For reference, a healthy EF is 52–74% [194]. In addition to the modification in EF, the Renin–Angiotensin–Aldosterone System (RAAS) activity is greatly increased within HF patients, which can result in decreased cerebral blood flow [195]. 

The reduction in cerebral blood flow within HF patients is often associated with cognitive decline, although the specifics seem to be influenced by the type of HF [196,197,198]. Other theories suggest that other factors such as increased levels of GFAP are responsible for cognitive changes [199,200,201]. Regardless of the mechanism, patients with HF have EEG abnormalities that closely resemble those of patients with cognitive impairments [202] and a 60% chance of developing dementia [35]. 

#### 2.2.6. Summary

Clinical assessment of primary CVD reveals a strong relationship between disruptions to the cardiovascular system, altered cerebral blood flow, and resultant cognitive impairment or psychological symptoms. Thus, for new and effective therapies to be established, disease models must reflect this complex interaction. Current models of NDDs often evaluate and focus on the CNS limiting the pool of knowledge surrounding interactions with the cardiovascular system. In the subsequent sections, we will summarize the current information regarding cardiovascular changes in traditional models of NDDs, CNS changes in models of CVD, and combinatorial models. 

## 3. Animal Studies 

### 3.1. Neurodegenerative Diseases

Within the published literature regarding animal models of NDDs, the focus of most studies concerns changes within the CNS. However, within these models, there have also been reports of changes to the cardiovascular system. In the following section, our goal will be to highlight the current understanding of changes to the cardiovascular system within these animal models of neurodegeneration. 

#### 3.1.1. Alzheimer’s Disease

The majority of AD research has focused on cardiovascular dysfunction as a risk factor and initial contributor to the development of disease, with only physical changes to the cerebral vasculature such as blood–brain barrier impairments, cerebral amyloid angiopathy, and reduced cerebral blood flow being commonly reported [203]. However, a few studies have suggested that the development of AD-related pathology may lead to cardiovascular impairments. Specifically, studies in the Tg2576 mouse model of CAA, which overexpresses the amyloid precursor protein (APP) Swedish (K607N/M671L) mutation, have shown that amyloid beta aggregates are present within the heart and that their presence was accompanied by fibrosis, expansion of the left ventricle and altered EF [204]. Similarly, work in the 5XFAD mouse model of AD, which overexpress APP with three familial AD mutations (K607N, M671L, and I716V) and double mutated PSEN1 (M146L/L286V), also supported the presence of a progressive disruption to the left ventricle EF which advanced with the severity of disease and was accompanied by cardiac fibrosis [205]. Additional work in APP/PS1 mice showed altered diastolic functioning, increased heart rate, cardiac fibrosis and hypertrophy alongside amyloid deposits within cardiac tissues [206,207]. Finally, research using the 3xTg mouse model reported decreased systolic functioning, increased heart rate, increased aortic stiffness/thickness, and elastin fragmentation in cardiac tissues [208]. While not fully elucidated, researchers speculate a variety of underlying mechanisms, including abnormal mitochondrial function, deposits of protein aggregates in cardiac tissue [206], broad autonomic dysfunction [207], and altered calcium homeostasis [208].

#### 3.1.2. Parkinson’s Disease

The prominence of autonomic dysfunction in PD patients has led researchers to focus investigations heavily on this aspect of disease in both toxin-based and genetic rodent models of PD. Despite this focus, a few studies have discussed physical changes in the cerebral vasculature as well as potential cell-autonomous changes in the heart.

##### Autonomic Dysfunction in Toxin Models of PD

One of the most common means of modelling PD in rodents involves the injection of a neurotoxin such as 1-Methyl-4-phenyl-1,2,3,6-tetrahydropyridine (MPTP), 6-hydroxydopamine (6-OHDA) or rotenone into deep brain nuclei to destroy the dopamine-producing neurons of the substantia nigra. Many studies have investigated autonomic-based changes of the cardiovascular system and the majority of these studies have indicated that either unilateral or bilateral injection of 6-OHDA or rotenone results in decreased heart rate variability [209,210,211,212], reduced heart rate [209,213], reduced mean arterial pressure at baseline [211,213,214], and abnormal baroreceptor and chemoreceptor reflexes [214,215]. In female rats, the effects on the cardiovascular system are milder, with no change in heart rate or heart rate variability [216,217] and a decrease in mean arterial pressure. Studies with female mice that have undergone an ovariectomy suggest that oestrogen is responsible for this protective effects [216]. Studies in mice involving injection of the toxin MPTP showed a decrease in mean and systolic blood pressure, increased heart rate, and altered heart rate variability [218]. In contrast to these general findings, some unilateral toxin models have reported increases in mean arterial pressure instead of a decrease [219,220]. While the exact reason for this difference is unclear, one study did report that the side of injection significantly affected the direction of the change in mean arterial pressure [220]. 

##### Autonomic Dysfunction in Transgenic Models of PD

In addition to chemical models, transgenic mouse models which overexpress human alpha-synuclein are also commonly used in PD research. To date, only a few studies exploring cardiovascular changes have been performed in these types of transgenic systems. Still, the majority have supported the same findings as the toxin model with some minor differences. Specifically, they observed elevated heart rate [221,222], altered autonomic control of heart rate [221], a reduced change in heart rate in response to stress [221], altered baroreceptor activity [222], and aggregated alpha-synuclein within cardiac tissue [223]. However, at least one study reported no changes to blood pressure or heart rate variability [224]. 

The reason for the reversal of the heart rate effect between transgenic and toxin models is not yet known, but the overall alignment of these two classes of models supports the idea that autonomic dysfunction is a core component of PD pathology. To date, no direct consensus on the origin of the autonomic disruption in models has been reached. Prominent theories include a disruption in the signalling of nuclei responsible for autonomic regulation of the heart [211,212,213] and loss or dysfunction of sympathetic innervation of the heart [218,225,226,227]. 

##### Cell Autonomous Disruption of Cardiomyocytes

The majority of studies aimed at the heart have focused on autonomic nervous system changes but there are some recent indications that chemical models of PD result in changes to lipid metabolism [228] and the response to oxidative stress [229] within the heart itself. These findings suggest that there may be disruptions to the cardiovascular system beyond the generally reported changes to autonomic regulation.

##### Blood–Brain Barrier and RAS Changes

Both blood–brain barrier integrity and cerebrovascular density have been evaluated in both transgenic and chemical models of PD. Collectively, these studies have described early changes to cerebrovascular density [230,231], augmented angiogenesis [232], and a disruption to the blood–brain barrier [231,232,233,234]. In all cases, these changes appeared either with or prior to motor impairments [230,231,232,233]. In chemical models, these changes tended to be static [231,232], while overall they progressed over time in transgenic mice [233], although this was not universally reported [234].

#### 3.1.3. Multiple Sclerosis

Similar to PD, autonomic dysfunction has been described as a component of MS pathophysiology [87,88,89,90], and has been the focus of most cardiovascular studies in multiple rodent models [235,236,237]. In addition to autonomic dysfunction other areas of research include assessment of the RAS, blood–brain barrier integrity, and morphological measures of the heart itself.

##### Autonomic Dysfunction in Rodent Models of MS

These studies consistently reported changes in ECG parameters, such as a longer QTc interval [235], decreased P wave duration [236], increased RR interval [237], increased heart rate [236,237], decreased left ventricle EF fraction, and decreased left ventricle fraction shortening [238]. Mechanistic investigations have predominantly focused on sympathetic nervous system receptors or changes to different ion channels that are involved with either the proper functioning of the heart, the brain, or the vagus nerve [237,239]. Together, these findings suggest that the expression of potassium channels and muscarinic acetylcholine receptors are dysregulated which may contribute to the reported autonomic changes [237,239].

##### Cell Autonomous Disruption of Cardiomyocytes

In addition to studies evaluating autonomic dysfunction, a few have reported cardiac changes such as altered cellular morphology [239,240], increased inflammation [239,240], hypertrophy, and fibrosis. Interestingly, sex-based assays indicated that fibrosis and hypertrophy but not inflammation were more apparent in male than female mice [238,239].

##### Blood–Brain Barrier and RAS Changes

Due to the inflammatory nature of MS and the known interactions between inflammation, RAS, and blood–brain barrier integrity, multiple studies have evaluated changes to these pathways in MS mouse models [241,242,243,244]. Studies of the RAS system demonstrate that immune cells express increased levels of the angiotensin I and IV receptors and that increased levels of angiotensin II and IV were present in the plasma of mice after induction of experimental autoimmune encephalomyelitis (EAE) [241,242,243]. Treatment with inhibitors of the two identified angiotensin receptors indicated that angiotensin I played a more significant role in brain pathology than angiotensin IV [241,242,243]. Assessment of blood–brain barrier integrity also demonstrated an inflammation-related disruption to zonulae occludens which followed the progression of disease in the EAE mouse model [244].

#### 3.1.4. Amyotrophic Lateral Sclerosis

Numerous studies using multiple mouse models have assessed blood–brain barrier disruptions in ALS, but few have addressed interactions with the cardiovascular system.

##### Autonomic Dysfunction in Rodent Models of ALS

To date, only three studies which used the same transgenic mouse model of ALS have measured autonomic dysfunction. Together, these studies have reported that the SOD1G93A mouse model displays elevated blood pressure [245] and increased heart rate [246] at early stages of disease, which gradually diminishes with disease progression. Based on immunohistochemical studies, the authors attribute these changes to disruptions of the autonomic nervous system, although no specific evaluation of innervation of the heart or heart rate variability was performed [247]. 

##### Blood–Brain Barrier/Blood–Spinal Cord Barrier and RAS Changes

Motor neurons are the primary neuronal population affected in ALS and, as such, the spinal cord is a focus of research in this area. As part of the CNS, the spinal cord contains a barrier between it and the blood, just like the brain. This barrier is termed the blood–spinal cord barrier, and it shares the features of the blood–brain barrier. Assessment of multiple mouse models of ALS supports that there are disruptions to both CNS barriers [248,249,250,251], although the specific changes differed slightly between models. For example, in the C9orf72 hexanucleotide repeat expansion mouse model, transporters of the blood–brain barrier were dysregulated while paracellular permeability remained unaffected, barring a minor decrease to zonula occludens-1 [248]. The prominence of disruptions to blood–brain barrier transporters was supported by work in G86R mSOD1 mice which showed upregulation of ATP-binding cassette transporters [249,250,252], as well as increased permeability to lipophilic marker [253]. Other reports in SOD1 transgenic models, which performed a more comprehensive evaluation of the blood–brain barrier, indicated that there was also impaired paracellular permeability [251,253,254]. This altered permeability was accompanied by reductions in tight-junction proteins such as occludin [251] as well as ultra-structural changes [254].

#### 3.1.5. Huntington’s Disease

The ubiquitous nature of HTT expression has led to a greater research focus on peripheral aspects of disease than what has been reported for most neurodegenerative disorders. Despite this increased focus, there is still little consensus on the relevance of peripheral phenotypes, with some studies suggesting strong correlations while others report no overall effect on disease severity [255,256]. Regarding the cardiovascular system, studies in multiple mouse models have reported a broad range of dysfunction including altered resting heart rate [220,257,258,259,260,261], reduced heart rate variability [257,261], reduced cardiac output [257,260,262,263,264,265,266,267], reduced left ventricle volume [257,263,266,268], reduced right ventricle volume [268], abnormal electrical activity in the heart [257,258,260,269], diastolic and systolic blood pressure [259], modulation of the RAS [127], impaired blood–brain barrier integrity [16,270,271,272], and cardiac fibrosis [257,263,264,265]. Currently, there are two predominant theories to explain cardiac involvement: one suggests that the presence of mHTT within the heart results in cell-autonomous disruption, while the other posits that disruption of the autonomic nervous system and the nuclei that regulate it results in cardiac impairments. The evidence supporting these theories is summarized below.

##### Cell Autonomous Disruption of Cardiomyocytes by mHTT in the Heart

The first indication that mutant huntingtin might play a role in cardiac dysfunction came from a study evaluating the effects of adding polyglutamine repeats to myosin [273]. While this did not directly impact huntingtin, the targeting of myosin made the manipulation specific to the heart. This specific expression of expanded CAG repeats was sufficient to lead to protein accumulation as well as signs of cardiac dysfunction [273]. This idea was further supported by one of the first studies of cardiac dysfunction in a transgenic mouse model of HD, which reported reduced heart weight, left ventricle dilation as well as mHTT aggregates within myocytes [263]. Finally, evidence of cell-autonomous dysfunction was generated in a study where genetic-reduction of mHTT specifically within the heart led to an amelioration of some cardiovascular symptoms [264]. However, the importance of mHTT has not been universally supported as at least one study failed to find evidence of aggregated mHTT within the heart, despite their presence within skeletal muscle [274]. Even in the absence of a clear role of aggregation, there is evidence that altered autophagic clearance [275], increased apoptosis [262,265,276], and altered mTORC1 signalling [277] occur within cardiomyocytes in different HD mouse models which support the idea of cell autonomous dysfunction within the heart.

##### Autonomic Dysfunction

More clinical studies have focused on the relevance of autonomic dysfunction to cardiovascular changes in HD than any other potential mechanism. This includes the evaluation of normal physiological means of adjusting to changes in systemic blood pressure, the baroreceptor reflex, as well as aspects of heart rate which are strongly linked to the autonomic nervous system such as heart rate variability. Studies in multiple mouse models support the presence of both altered baroreceptor reflex as measured by response to pharmacological modification of blood pressure [257,259] and heart rate variability [257,261]. Additionally, nuclei associated with the autonomic nervous system were shown to be hyperactive in HD mice models [258] and restoring levels of brain-derived neurotrophic factor (BDNF) or signalling through ryanodine receptors in sympathetic nuclei has been shown to ameliorate changes to heart rate and/or altered electrical activity in mouse models of HD [221,278].

##### Blood–Brain Barrier and RAS Changes 

Relatively fewer studies have evaluated the blood–brain barrier and RAS in HD mice. Reports of changes to the blood–brain barrier [16,270,271,272], indicate that disruptions to the barrier integrity appear notably early within a particularly severe transgenic mouse model of HD [271]. In contrast to this, one study of peripheral RAS in a mouse model of HD reported no changes to plasma levels [127].

#### 3.1.6. Summary

The frequent evidence of cardiovascular dysfunction in different animal models of NDDs provides evidence that there are either shared aetiologies of cardiovascular and ND diseases or that NDDs are sufficient to cause some disruptions to the cardiovascular system. These include but are not limited to, autonomic dysfunction, increased permeability of the blood–brain barrier, and changes to the RAS. Mechanistically, changes to the integrity of the blood–brain barrier manifest as the deterioration of pivotal cell types such as pericytes and endothelial cells, cerebral blood flow reductions that hinder protein aggregate clearance, and vascular inflammatory phenotypes stemming from glial cell activation and immune cell invasion [279]. Concerning the RAS, disease pathology likely stems from the pro-inflammatory and pro-oxidant mechanisms of its pressor axis, the coupling of Angiotensin II and the Angiotensin 1 receptor [77]. In this, excessive activation has been shown to alter neurotransmitter activity [280], generate high levels of reactive oxygen species through the NADPH-oxidase complex [77], and contribute to disease-specific protein aggregation [281,282]. 

Clinical evidence suggests that CVD frequently precedes the onset of behavioural phenotypes and, as such, a full recapitulation of disease phenotypes may require the combination of cardiovascular models with more traditional models of NDDs.

### 3.2. Cardiovascular Disease 

The link between the cardiovascular system and the CNS has been well studied in models of cardiovascular risk factors, such as AS and hypertension. In these cases, differences are clear and pronounced. In other conditions, research is more sparse and the summary below is restricted to models where neurological deficits have been described.

#### 3.2.1. Hypertension and Atherosclerosis 

Hypertension, otherwise known as high blood pressure, is one of the most studied conditions when researching cardiovascular dysfunction due to its high prevalence in the population. This research focus has led to the development of a vast range of animal models which target the heart and/or kidneys using mechanical and/or hormonal stress pathways. Mechanically, arteries are narrowed or occluded to place strain on the cardiovascular, whereas hormonal stress takes advantage of the global RAS. The RAS is a hormonal pathway which is largely implicated in control over the cardiovascular system, specifically in managing fluid volume, neuronal endocrine function, electrolyte balance, and most importantly, blood pressure [283]. In short, the RAS involves the release of the renin enzyme from the kidneys into the bloodstream, which cleaves angiotensinogen, converting it into angiotensin I, which is then converted into angiotensin II by the angiotensin-converting enzyme (ACE) [284] (Figure 1). 

##### Surgical Models

Surgically, multiple intervention strategies are available. To begin, the Transverse Aortic Constriction (TAC) procedure is a surgically induced model of cardiac pressure overload-induced hypertrophy, hypertension, and gradual heart failure [285]. This procedure involves reducing the diameter of the aortic arch by threading a ligature around the aorta between the innominate and left common carotid artery and tightening around a small gauge needle. This model has been shown to increase blood pressure artificially in the downstream vasculature and has also been noted to cause hypoperfusion of the kidneys, altering the expression of renin within the cardiovascular system [286] (Figure 1). 

The kidney clip procedure is another prominent surgical model used to exemplify animal hypertension. Devised initially by Goldblatt and colleagues in the early 20th century, the kidney clip procedure involves placing a metal clip(s) on the renal artery which occludes blood flow and produces hypoperfusion of the kidney and thereby alters the downstream expression of renin renovascular hypertension (Figure 1) [287]. Specifically, the procedure can be performed in one of three ways; the two-kidney one-clip (2K1C) procedure, the lesser used two-kidney two-clip model (2K2C), and the one-kidney one-clip model (1K1C) [288,289]. The 2K1C procedure involves a single clip placed on the renal artery. In contrast, the 2K2C method involves placing an additional clip on the contralateral renal artery (Figure 1). Both methods activate the RAS, but 2K2C more frequently leads to sustained hypertension, due to technical challenges which may result from intersubject differences in the diameter between kidneys [290]. Finally, the 1K1C model involves the removal of one kidney and placing a clip on the contralateral renal artery, which halts renin secretion and, subsequently, downstream angiotensin synthesis [289]. 

Lastly, the Angiotensin II osmotic pump is a highly favoured model of surgically induced systolic hypertension. This procedure involves the surgical implantation of a subcutaneous osmotic pump device either into the back of the animal, posterior to the scapulae, or directly cannulated into the cerebral ventricles which continuously infuses angiotensin II, the main effector molecule in the RAS, at a constant rate over a short time course [291] (Figure 1).

##### Genetic Models

Here, we describe the spontaneously hypertensive rat (SHR), an inbred strain of the Wistar Kyoto rat and the high blood pressure (BPH/2) mouse model, both selected for high blood pressure phenotypes. The SHR model gradually develops severe hypertension by 2 to 4 months of age, with blood pressure stabilizing by six months [292]. While the full genetic basis of the development of hypertension in the SHR model is still debated due to its complex polygenic nature, most researchers agree there is some manipulation of the genes related to the RAS [293,294]. Additionally, while considered a low-renin model of hypertension, the SHR model demonstrates increased angiotensinogen production, subsequently increasing circulating angiotensin II [295] (Figure 1). The BPH/2 mouse model demonstrates increased systolic blood pressure compared to their normotensive counterpart (Blood Pressure Normal—BPN/3J) by three months of age [296], partly due to neurogenic mechanisms controlling renal renin secretion [297]. 

##### Dietary Models

Beyond surgical and genetic manipulations, less invasive and costly methods have been developed to induce cardiovascular dysfunction. The most common method used with murine models is dietary supplementation with salt. Otherwise referred to as a high-salt diet (HSD), this method allows researchers to study hypertension and its systemic effects on the entire bodily system in a non-invasive fashion. In this model, animals are fed an excess of NaCl within their regular diet, often between 4–8%, for lengths of time ranging from a few days to months. Hypertension reportedly develops after six weeks in rats [298] and eight weeks in mice [299]. However, there remains some debate in the scientific community regarding the efficacy of a high-salt diet in generating hypertension. 

##### Neurological Effects

Evaluation of the CNS in these different model systems demonstrates a fairly consistent effect on neuropathology despite the different mechanisms of action (Table 1). Hallmark changes include cognitive impairments, reduced tight-junction protein expression, greater blood–brain barrier permeability, increased ventricle volume, and activation of glial cells (Table 1). The most variable finding was cerebral blood flow which was consistently changed but with different directionality depending on the experiment and the region evaluated (Table 1). Overall, the consistency of these reports supports an interaction between hypertension and changes to the CNS regardless of the specific mechanism employed.

#### 3.2.2. Metabolic Disease, Coronary Artery Disease and Atherosclerosis 

Metabolic disease, or syndrome (MetS), refers to concomitant pathologies/risk factors for coronary artery disease (CAD), including obesity, insulin resistance, type 2 diabetes, dyslipidaemia, and hypertension [325]. Encompassed herein, CAD refers to reduced blood flow to cardiac tissues as a result of AS. Various animal models, particularly mouse models, have been used extensively across the literature to target the interplay between MetS, CAD, and AS via the exploitation of relevant dietary and genetic factors. 

Dietary models use high-fat or high-sugar supplementation, or a combination thereof, to increase the accumulation of lipids within vessels. On the other hand, genetic models help to overcome mice’s overall resistance to the progression and development of CAD by introducing predisposing genetic mutations such as *ApoE* and low-density lipoprotein receptors (*Ldlr*) or manipulating hunger-signalling, as seen in leptin knockout models. 

##### Dietary Models

Similarly to the HSD model utilized in hypertension research, dietary models are commonly used intervention strategies to study MetS and corresponding CAD. A high-fat diet (HFD) is used to induce obesity and consists of dietary supplementation with chow supplemented with lard, milk fats, and various vegetable oils such that 40–60% of caloric energy is obtained from fats [326]. Generally, these animals demonstrate a 10–20% increase in body weight over standard chow controls, with differences beginning as early as two weeks of intervention and compiling gradually over a prolonged time period [326,327]. Other features of this model include insulin resistance [328], hyperglycaemia, hyperinsulinemia, and hypertriglyceridemia [326].

A high-sugar diet, also referred to as a high-sucrose or high-fructose diet, is similarly used to study MetS and CAD pathologies. Unlike HFD, the high-sugar diet is a non-obesity model and involves dietary supplementation, typically between 60–70% of caloric energy, from fructose or sucrose either in the diet or drinking water [329]. Other features of this model include insulin resistance and dyslipidaemia [329]. A combination high-fat high-sugar diet (HFHS) or “Western diet” consists of dietary supplementation with both fat and sugar, at ~36% of energy from sugar and fat, respectively, [330]. These animals display increased body weight, glucose intolerance, and insulin resistance [331]. 

Finally, the atherogenic diet is used to model atherosclerosis in genetically susceptible strains of mice and rats [332]. This diet supplements standard chow with 15.8% fat content, 1.25% cholesterol, and 0.5% sodium cholate. The diet is typically maintained over several months, with aortic atherosclerotic lesions appearing following 3.5 months of intervention with increasing severity when prolonged to 9–12 months [332,333]. 

##### Genetic Models

When studying MetS and the corresponding CAD, multiple transgenic models exist, including Leptin-deficient (*Lep ob/ob*), Leptin receptor-deficient (*LepR db*/*db*) mice Low-density lipoprotein receptor knockout (*Ldlr*−/−), and Apolipoprotein E-deficient (*ApoE*−/−) mice. 

The leptin protein has long been associated with obesity phenotypes due to its role in long-term energy homeostasis [334] and atherogenesis [335]. The *Lep ob/ob* mouse model is a genetic model of MetS backcrossed to the C57BL/6J strain that is homozygous for a spontaneous obesity mutation in the Leptin (*Lep*) gene leading to a deficiency in the functional leptin protein [336]. These mice are characterized by morbid obesity, with moderate obesity seen as early as 4 weeks of age, and diabetes-like pathology, including hyperglycaemia, elevated insulin, and glucose intolerance [337]. Similar to the *Lep ob/ob* mouse, the Leptin receptor-deficient (*LepR db/db*) is another genetic model of MetS with a C57BL/6J background with a homozygous mutation to the leptin receptor gene leading to a defective leptin receptor [336]. Likewise, these mice also demonstrate obese phenotypes at 3–4 weeks of age [338]. Both models are known to be resistant to AS, not developing atherosclerotic lesions spontaneously [339] and generating only small lesions when the phenotype is compounded with CVD or ApoE knockout. 

The low-density lipoprotein receptor (*Ldlr*−/−) model is another genetic knockout model commonly utilized to assess familial hypocholesterolaemia as a result of the increase in circulating low-density lipoprotein due to impaired clearance associated with the absence of the receptor [340]. As such, the phenotypes induced through the Ldlr−/− model largely depend on the diet used. Similarly to the leptin deficient models, these mice have been shown to be resistant to atherosclerosis, however, when placed on an atherogenic diet, they develop aortic atherosclerotic plaques [341]. 

Finally, the Apolipoprotein E-deficient (*ApoE*−/−) is characterized by increased levels of atherogenic cholesterol-rich lipoproteins due to poor lipoprotein clearance, similar to the *Ldlr*−/− model. Phenotypically, these mice display increased plasma cholesterol levels and develop severe atherosclerotic lesions as early as ten weeks of age and earlier, at eight weeks, with supplementation dietary intervention [342].

##### Neurological Effects

When CNS effects are assayed the various models of AS universally display increased memory impairments which were accompanied by increased blood–brain barrier permeability and decreased tight junction protein (Table 2). Beyond these consistent findings, individual reports described more severe pathology including a decrease in total brain volume and neuronal density, increased astrocyte activation, and greater levels of pathological forms of amyloid-beta and tau (Table 2).

#### 3.2.3. Myocardial Infarction and Heart Failure

While a variety of models exist, only one, to the knowledge of the authors, has been evaluated for neurological changes. This model, the MI model of heart attack and HF, involves surgical ligation of the left anterior descending artery which effectively stops blood flow to the inferior muscle and causes tissue necrosis [376]. The general result of this MI model is the induction of mechanical stress on the cardiovascular system which leads to greater inflammation, increased levels of circulating angiotensin II, cardiac hypertrophy, and gradual heart failure [377,378]. 

Behaviourally, MI rats have been reported to demonstrate anxiety-like behaviour [379], depression-like behaviour [380], and long-term memory impairments [381,382]. These changes were associated with pathological changes in the CNS including, reduced cerebral blood flow [383], increased migration of glial cells into blood vessels [384], greater activation of microglia and astrocytes [379], and decreased dendritic length and spine density [381,382]. 

#### 3.2.4. Atrial Fibrillation

Numerous models of AF exist, including SHR, mice with genetic knockouts of cardiac-specific liver kinase 1 and angiotensin infusion to name a few [385], it can also be modelled by exposing mice to the chronic unpredictable mild stress model of depression, which suggests that dysfunction of the nervous system may be sufficient to cause this form of CVD [386]. The importance of the brain in the development of some forms of AF is further supported by studies reporting that blockade of the angiotensin II type 1 receptor specifically in the brain is sufficient to prevent AF in the SHR model [387]. Although the aforementioned studies support the effect of brain pathology on AF, little is known about the impact of AF on the brain.

#### 3.2.5. Cardiomyopathies

There are many types of CMs, some of which are acquired while others are the result of genetic mutations [388]. In addition to their causes, myopathies can be further subdivided into primary and secondary CMs based on the degree of involvement of other bodily systems [388]. Secondary CMs, including those resulting from mitochondrial mutations, often include neurological features as a core component of the pathology and as such will not be discussed in this review. Instead, the discussion will be focused on the most common primary CMs for which interactions with the CNS have been studied in mice: dilated CM and hypertrophic CM [388].

The Tg9 genetic model of dilated CM overexpresses long-chain acyl-CoA synthetase and is characterized by key features of human CM such as cardiomegaly, progressive fibrosis, and reduced left ventricle EF [389]. To the knowledge of the authors, no studies have evaluated behavioural changes or alterations in cerebral blood flow, however, one study reported decreased sympathetic innervation of the heart [389]. 

KI-TnC-A8V^+/+^ mice have a mutation inserted into endogenous troponin C to model hypertrophic CM. At 4 months of age, these mice display abnormalities on echocardiograms including higher ventricular volume and a higher EF which progress to more severe disruptions including bradycardia and autonomic dysfunction by 7 months of age. Evaluation of behavioural phenotypes in this model revealed increased anxiety- and depression-like behaviour which correlated with altered parasympathetic nervous system activation [390]. In addition to the altered innervation, reductions in the prefrontal cortex and caudate putamen volumes were reported [390]. These findings suggest that there may be interplay between cardiomyopathies and the nervous system, although further research is required to fully elucidate this relationship.

#### 3.2.6. Summary

The study of CNS changes after cardiovascular dysfunction is relatively rare, however, the existing literature demonstrates a complex and sometimes bi-directional interplay between CVD and the CNS. The model-based differences in the neuropathology and behavioural changes highlight the importance of selecting the appropriate cardiovascular model for combinatorial studies.

### 3.3. Combinatorial Models

Combinatorial methods that incorporate CVD and NDD models are crucial to furthering our understanding of how these pathologies intersect. Despite this, studies within this domain are limited, many of which focused on cardiovascular dysfunction within AD, with a smaller fraction focusing on PD, MS, ALS, and HD. Nonetheless, evidence suggests the coalescence of models of CVD and NDD worsens disease-specific pathology bidirectionally. 

#### 3.3.1. Myocardial Infarction

Within the MI model, researchers have examined its interaction with murine models of AD, specifically the APP/PS1 mouse model, a double transgenic model expressing human APP and PSEN1. Therewithin, MI was induced at 20 weeks, followed by assessment 4 weeks post-surgery, at which time APP/PS1/MI mice demonstrated increased amyloid-beta plaques and phosphorylated tau within the hippocampus [391]. Behaviourally, memory impairments were seen in the APP/PS1/MI mice, demonstrated as increased escape latency in the Morris water maze and increased anxiety-like behaviour reported as increased time in an aversive environment of the plus-maze discriminative avoidance task [391]. 

Beyond AD, Xu and colleagues (2020) utilized the MPTP model of Parkinson’s disease, following mice at 3- and 9-months post-MI. However, researchers found no differences in terms of glial cell activation, alpha-synuclein aggregation or behavioural impairments between the MI and wildtype mice [392]. 

#### 3.3.2. Transverse Aortic Constriction

De Montgolfier and colleagues investigated TAC in combination with the APP/PS1 mouse model of AD, described above. Six-weeks post-surgery, they reported an increase in astrocyte activation, amyloid deposition, and cellular senescence in the ipsilateral cortical tissue in 6-month-old animals compared to all other experimental cohorts [301]. Behaviourally, the TAC-APP mice demonstrated minor memory impairments as measured by a slight delay in latency to reach the hidden platform and reduced time in the target quadrant of the Morris water maze task [301]. 

#### 3.3.3. Angiotensin II Osmotic Pump

The same APP/PS1 mouse model has also been tested in combination with implantation of the angiotensin II osmotic pump at 10 months of age. In these mice, 1 month of chronic angiotensin II administration led to reduced cortical cerebral blood flow and impaired acquisition of the Morris water maze task within the four-day training period, demonstrating impaired spatial learning abilities [393]. A second angiotensin study using the 5XFAD mouse model of AD found that at 12 months of age, a similar impairment in spatial learning abilities, as measured by increased latency to escape in the Morris water maze. Evaluation of post-mortem tissue further demonstrated an increased activation of microglia in the hippocampal tissue and amyloid deposits in cortical cerebral arteries following 1 month of angiotensin II administration [394]. Finally, the TgAPP21 rat model of AD, which expresses pathogenic APP, displayed increased microglia activation within the corpus callosum, cingulum, internal capsule, and hippocampus following 2 months of angiotensin II administration at approximately 10 months of age [312].

#### 3.3.4. Kidney Clip

Investigation of the 2K1C method of renovascular hypertension within the 3xTg mouse model of AD, a triple transgenic model expressing mutations in APP, Tau, and PSEN1 genes, displayed similar results to the Angiotensin II experiments described in the previous section. These mice demonstrated increased deposition of amyloid-beta, enhanced disruption of the blood–brain barrier, and microglial activation detected in the hippocampal tissue 1-month post-surgery at all ages examined (2, 5, and 7 months) [395]. Additionally, researchers observed increased levels of phosphorylated tau in the striatal tissue at all observed surgical time points [395]. Finally, the 2K1C surgery led to precipitation of cognitive deficits, as measured by decreased exploration of the novel object in the novel object recognition tasks at the 2 and 5-month timepoints [395].

#### 3.3.5. High-Salt Diet

HSD has been tested in two independent experiments in APP/PS1 mice and one experiment in the PLP-hαSYN model of synucleinopathy. The first of these studies initiated feeding at 2-months of age and tracked mice for 3 months. At the end of this period, they reported that the diet augmented cerebral blood flow in both APP/PS1 and WT mice, but specifically induced hypertensive phenotypes in WT mice. When AD model specific traits such as brain volume and amyloid-beta accumulation were assessed, no change and a decrease in the number of plaques was observed, respectively. These findings contrast with the second study where increased accumulation of amyloid-beta and tau were reported in addition to increased cognitive impairment, greater microglial coverage, reduced pericyte interaction with blood vessels, and blood–brain barrier impairments [279]. The largest difference between these two studies was the age of the intervention with the second beginning 4 months later at 6 months and lasting 4 months instead of 2 months [279]. The increased age of the mice may have influenced the response to the high-salt regimen. Evaluation of a high-salt diet in a model of alpha-synucleinopathies (PLP-hαSYN) resulted in very different findings with no differences in terms of microglia/astrocyte activation, neuroinflammation, alpha-synuclein aggregation or behaviour being detected [396]. 

#### 3.3.6. High-Fat Diet

Implementation of HFD is one of the most common methods of assessing the effects of cardiovascular disease on NDD having been evaluated in mouse models of AD, PD, HD, ALS, and MS. In AD, two different mouse models have been evaluated, the 3xTg and 5XFAD. Specifically, 3-month old 3xTg mice were fed HFD for 12 months which incited increased CAA at 6 months and a significant increase in serum cholesterol as well as atherosclerotic plaque area in the thoracic-abdominal aorta and amyloid-beta micro clots within cerebrovascular vessels at 9-months [397]. After 12 months of HFD 3xTg mice displayed increased blood–brain barrier permeability, more reactive astrocytes within the hippocampus, and extensive tau hyperphosphorylation [397]. In another study undertaken in the same 3xTg mouse model, mice were fed HFD from 1 month until 5 months of age. Memory impairments, as indicated by higher escape latency during training and reduced time in the target quadrant in the Morris water maze, were detected as a result of HFD [398]. Similarly, a 10-week HFD intervention in 13-month-old 5XFAD mice reportedly displayed increased amyloid-beta deposits in hippocampal vasculature and the middle cerebral artery [399]. Additionally, researchers discovered decreased expression of the tight junction protein occludin, suggesting increased blood–brain barrier permeability [399]. Behaviourally, these mice demonstrated increased escape latency and reduced time in the target quadrant as seen in the previously described 3xTg model. 

In models of PD, results were less consistent with a 5-week HFD intervention after 6-OHDA infusion leading to increased dopamine depletion within the substantia nigra and striatum [400] whereas a very high-fat diet treatment (ketogenic; 90% fat) across an 8 week period in the MPTP model of PD, attenuated motor symptoms, decreased glial cell reactivity and restored dopaminergic neuron loss [401]. These differences may be reflective of the diet used as placing the BACHD model of HD on a similar ketogenic diet (77% fat) for 3 months led to amelioration of circadian dysfunctions and improved performance on the rotarod and balance beam tests [402]. 

In the G86R SOD1 model of ALS, a milder HFD (21% butter fat supplementation) beginning at 6 weeks of age and continuing until humane endpoints were reached, demonstrated an increased average lifespan of HFD fed mice [403]. 

Finally, in the EAE model of MS, HFD instituted for 8 weeks reportedly caused earlier disease onset and severity in these mice [404]. Additionally demonstrated increased spinal cord lesions and necrosis, increased blood–brain barrier permeability, and activated microglia [404]. 

#### 3.3.7. High-Sugar Diet

Implementation of a high-sugar diet is a lesser-studied intervention in combination with neurodegenerative models. Nonetheless, research using 2-month-old APP/PS1 mice supplemented with full sugar chocolate pudding across an 8-week time course demonstrated globally increased cerebral amyloid-beta plaques and demonstrated memory impairments in the novel object recognition test [405]. In another study utilizing the same model beginning at 10 weeks of age, these mice demonstrated hindered daily living as measured via the food pellet burrowing paradigm as well as increased cortical amyloid-beta and reactive astrocytes following 28 weeks of high sugar diet (67.3% carbohydrates) [406]. 

#### 3.3.8. Western Diet (High-Fat High-Sugar)

The more variable western, or high-fat high-sugar, dietary intervention has been used finitely in combination with models of neurodegeneration. Specifically, the late onset Alzheimer’s disease (LOAD1) mouse model of AD, a transgenic model double homozygous for *APOE4* and *Trem2* R47H mutation additionally combined with the M28L variant of phospholipase C Gamma 2 (*Plcg2*) or the 677C > T variant in methylenetetrahydrofolate reductase (*Mthfr*), was fed with a 45% fat and 36% carbohydrate diet for 10 months, beginning at 2 months of age. Herein, researchers found increased microglia activation in the female LOAD1/*plcg2* mouse cortex [407]. In a secondary study utilizing the 3xTg mouse model under the Western diet for 14 weeks beginning at 7 months of age, mice were found to have reduced glucose tolerance and behaviourally demonstrated significant deficits in spatial memory illustrated by reduced alternation percentage and fewer entries into novel arms within the Y-maze paradigm [408]. Finally, in the McGill-R-Thy1-APP heterozygous rat model of AD, expressing human APP with three familial mutations (K607N, M671L, and I716V), beginning at 1 month of age and following 6.5 months of intervention demonstrated increased systemic insulin levels and increased accumulation of amyloid-beta in hippocampal tissue [409]. Moreover, these mice displayed worsened spatial memory via decreased alternation in the Y-maze paradigm and higher escape latencies in the Morris water maze [409]. Beyond AD, one study examined the western diet (47% fat; 38% carbohydrates) in combination with the G93A SOD1 model of ALS beginning at 6 weeks of age and found significantly increased survival time (90%) and increased delay between disease onset and death [410]. 

#### 3.3.9. Atherogenic Diet

Implementation of the atherogenic diet has been researched in combination with the Tg2576 mouse model of AD, a model that overexpresses mutant APP with two familial mutations (K607N/M671L). Beginning between 7–9 months of age, mice underwent 4 months of dietary intervention which resulted in significantly increased aortic AS and amyloid-beta plaques within the hippocampus and cortex, showing a twofold increase in cerebral amyloid burden with the atherogenic mice [411]. Behaviourally, spatial learning impairments were demonstrated as increased escape latency in the Morris water maze [411]. 

#### 3.3.10. Leptin Deficiency (Lep ob/ob; LepR db/db)

Concerning leptin-deficient models of MetS, few studies have investigated the interplay with neurodegeneration, and to the best knowledge of the authors, the existing literature exclusively utilizes the *Lep ob/ob* model. Takeda and colleagues (2010) generated a double transgenic APP+/ob model to undertake their investigation and found that within 8 weeks, these mice displayed marked phenotypes of hyperglycaemia, hyperinsulinemia, and glucose intolerance as compared to the APP+ cohort [412]. Moreover, they found a significant increase in amyloid-beta 40 within cerebral micro-vessels beginning at 6 months of age and reactive astrocytes as well as significant total brain atrophy at 12-months in the APP+/ob mouse [412]. Finally, behavioural testing revealed severe deficits in learning of the hidden platform test within the Morris water maze at 8 weeks which was sustained to 12 weeks in the APP+/ob mice [412]. Beyond AD, one study was performed using a generated SOD1/ob double transgenic mouse to model ALS and findings suggest improved energy homeostasis and overall slowed disease progression in the model [413].

#### 3.3.11. Low-Density Lipoprotein Receptor (LdlR−/−)

Deficiency of the low-density lipoprotein receptor (*LdlR*−/−) has been investigated in combination with various AD models. One such study used the 5XFAD model crossed into the *LdlR*−/− background to generate a double transgenic mouse. In these mice, significant increases in amyloid deposits were reported in the hippocampus and cortex, as well as a global increase in ApoE protein within the brain at 4 months [414]. In contrast, these mice displayed decreased levels of reactive astrocytes and microglia in the cortex and hippocampus as compared to the 5XFAD cohort at the same 4-month timepoint [414]. Another study utilizing the Tg3576 mouse crossed with the *LdlR*−/− background found these mice had a twofold increase in cerebral amyloid burden as compared to the APP/LdlR+ cohort beginning at 11 months and progressing in severity to 24 months [415]. Behaviourally, these mice showed decreased anxiety-like behaviour in the elevated plus maze but deficits in learning through impairments in re-acquiring hidden platform paradigm in the Morris water maze at 13 months [415].

#### 3.3.12. Apolipoprotein E (ApoE−/−)

Finally, the Apolipoprotein knockout model (*ApoE*−/−) has been investigated in multiple models of neurodegeneration. First, the APP model with the V717F mutation was crossed with the *ApoE*−/− background and disease progression was studied at 9, 15, and 22 months of age. Herein, researchers found reduced amyloid deposition alongside reduced glial cell activation up until the 22 month timepoint [416]. Similarly, a second study in the APP V717F model of AD observed that amyloid-beta plaques could be found at both 12 and 15 months within the hippocampus in the ApoE deficient mice [417], but very few neuritic plaques were detected at 12 months or 15 months of age [417]. 

Beyond AD, the A30P model of PD crossed with the ApoE−/− background was found to attenuate alpha-synuclein-based neurodegenerative phenotypes, increasing the level of soluble alpha-synuclein protein, significantly delaying the onset of motor symptoms and improving overall median survival of these mice as compared to A30P/ApoE wildtype mice [418]. Similarly, EAE model of MS combined with the ApoE knockout background also exhibited delayed disease onset and reduced severity of clinical disease for the 10–29 day duration of the study [419]. 

## 4. Discussion

Upon reviewing the literature, there is a strong degree of overlap between cardiovascular and NDD rodent models with the majority of NDD models displaying cardiovascular changes and some evidence of central nervous system changes in models of CVD. Most of these interactions are currently thought to be mediated through a small number of mechanisms, including central disruption of the autonomic nervous system, increased protein accumulation within cardiomyocytes, changes to peripheral RAS, and disruptions to cerebral blood flow. However, it should also be noted that the central nervous system and the cardiovascular system share some overlap in properties, such as susceptibility to mitochondrial dysfunction and inflammation that may also underly some of the interactions between these disorders. Evidence for this comes from studies reporting that MS, PD, and AD all share molecular mechanisms or unique genetic risk factors with various CVD. In the case of PD, these relate to mitochondrial function [73], in AD they relate to APP processing [47] while in MS they relate to lipid transport and metabolism [86].

These shared risk-factors may help explain why, despite the presence of many shared pathological changes, there are still disease-related differences in the interactions between CVD and NDD. For example, both clinical and pre-clinical work in AD indicates that CVD and CVD risk factors are likely an early and potentially causal contributor to disease development which may be exacerbated by the development of protein pathology within the heart. In contrast, there is substantial evidence that induction of PD and MS is sufficient to induce changes to autonomic innervation of the heart with much less data to support an early contribution of CVD to the development of these two disorders. HD and ALS sit between these two extremes, with evidence of cardiovascular risk factors and cardiovascular changes including blood–brain barrier disruption appearing early in the disease and influencing age of onset as well as brain-mediated changes to the autonomic innervation of the heart and accumulation of misfolded proteins. While current evidence indicates that different mechanisms are at play in different NDD and CVD, care should be taken as this may be at least partially the result of the specific tests that are most commonly used in each disorder. With the focus on autonomic dysfunction in PD, few studies have evaluated the heart itself for indications of dysfunction. Similarly, no assessment of the autonomic nervous system has been reported to the best of the authors’ knowledge, in rodent models of AD. In cardiovascular disorders, studies evaluating behaviour tend to focus on single tests designed to measure specific factors, such as cognition or anxiety, leaving it unclear whether AF and CM solely lead to depression and anxiety.

Despite this caveat, combinatorial studies support a model where the disease context influences the effect of cardiovascular interventions. For example, HSD led to an increased accumulation of amyloid beta and tau in the APP/PS1 mouse model, but no accumulation of alpha-synuclein was apparent in PLP-hαSYN mice. Additionally, MI surgery led to an overall exacerbation of disease in AD mice but had no effect on the MPTP model of PD, while a Western diet was again detrimental in AD but beneficial in an ALS mouse model. While experimental differences may have contributed to these variable findings, this supports a disease-specific interaction between cardiovascular disease and neurodegeneration. 

## 5. Conclusions

Both NDD and CVD models display cardiovascular and neurological changes at baseline, reflecting the co-morbidities and shared risk factors previously seen in clinical studies. However, these model systems do not fully reflect clinical conditions unless combinatorial models are used. Current research suggests that combinatorial models can enhance disease-specific phenotypes and may be a powerful tool to better model clinical disease.

## 6. Future Directions

To fully understand the interactions between NDD and CVD, more extensive testing of combinatorial models across a range of disorders is required, along with a more in depth investigation into the influence of CVD on the brain in the absence of NDD susceptibility factors. Only after establishing a baseline understanding of how CVD impacts the brain will studies of disease-specific interactions allow for a meaningful understanding of disease-specific versus generic relationships.

## Figures and Tables

**Figure 1 biology-13-00764-f001:**
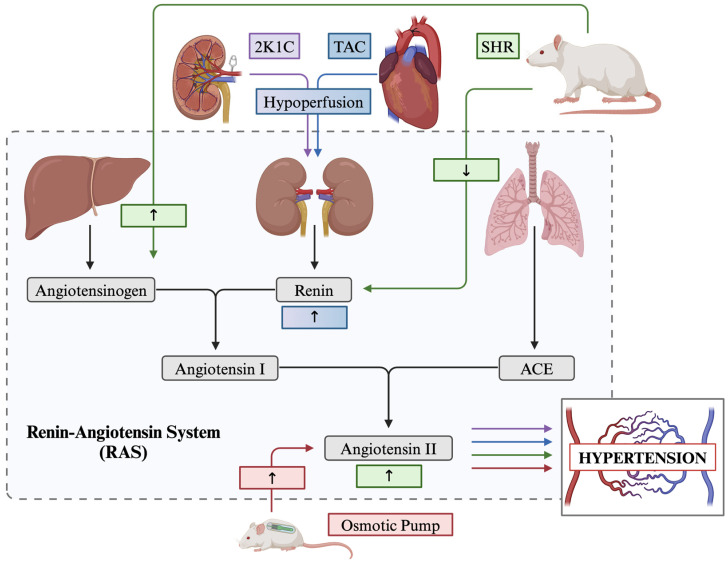
Overview of the renin–angiotensin system (RAS) and how described models of hypertension are associated across the pathway. 2K1C 2 kidney-1 clip; TAC transverse aortic constriction; SHR spontaneously hypertensive rat; ACE angiotensin converting enzyme.

**Table 1 biology-13-00764-t001:** Overview of features of cardiovascular disease (CVD), neuropathology, and behavioural deficits in reviewed murine models of hypertension (HT).

Disease	Model	Mech.	Features of CVD	Neuropathology	Behaviour
HT	TAC	S	Cardiac pressure overload-induced hypertrophy, HF [285]	↓ CBF/TJP expression, ↑ BBB permeability [300,301]	↑ Memory impairments [301,302]
HT	Kidney Clip	S	Renovascular HT [288]	↑ Ventricle volume, WM abnormalities [303], altered CBF & ↓ TJP expression [304]	↑ Memory impairments, ↓ Cognitive flexibility [304,305,306]
HT	Ang II Osmotic Pump	S	Systolic HT [307]	↑ Ventricle volume/BBB permeability, ↓ TJP expression [308,309,310], ↑ Glial cell activation [311,312,313]	↑ Memory impairments/Anxiety-like behaviour [312,314]
HT	SHR	G	Systolic HT [292]	↑ Ventricle volume, ↓ Grey matter [315], ↑ Astrocyte activation/BBB permeability [316,317]	Hyperactivity [318], ↑ Memory impairments [289]
HT	BPH/2	G	Neurogenic HT [297]	↑ Microglial activation [296]	↑ Spatial/Working memory impairments [296]
HT	High-Salt	D	Systolic HT * [319]	Altered CBF [320,321,322], ↑ BBB permeability [279]	Altered CBF [279,323], ↑ BBB permeability [324]

* Induction of systolic hypertension currently under debate in the scientific literature. ↑ and ↓ denote an increase or decrease respectively; S surgical intervention, G genetic intervention; D dietary intervention; HF heart failure; CBF cerebral blood flow; TJP tight junction proteins; BBB blood–brain barrier; WM white matter.

**Table 2 biology-13-00764-t002:** Overview of features of cardiovascular disease (CVD), neuropathology, and behavioural deficits in reviewed murine models of MetS, coronary artery disease (CAD) and atherosclerosis (AS).

Disease	Model	Mech.	Features of CVD	Neuropathology	Behaviour
MetS	High-Fat	D	Systolic HT, Cardiac hypertrophy, fibrosis & stiffness [343,344,345]	↓ TJP expression/Neuronal density [346], ↑ Astrocyte activation [347]	↑ Memory impairments [348,349]
MetS	High-Sugar	D	Systolic HT * [350], Cardiac fibrosis & Cardiomyopathy [87]	↑ Amyloid-beta & Phosphorylated tau [351]	↑ Spatial memory impairments [351]
MetS	High-Fat/Sugar	D	Cardiac fibrosis [352]	↑ BBB permeability/Astrocyte activation, ↓ TJP expression [353]	↑ Memory impairments [353]
CAD	Leb (ob/ob)	G	↓ Cardiac efficiency, cardiac hypertrophy [354]	↓ Total brain volume [355]/Neuronal density [356]	↑ Memory impairments [355]
CAD	Leb (db/db)	G	↓ Cardiac efficiency [354], Impaired angiogenesis [357] & ↑ Mean arterial pressure [358]	↑ BBB permeability [359]/. Phosphorylated tau [360]	↑ Memory impairments [361]/Depression-like behaviour, ↓ Anxiety-like behaviour/Pre-pulse inhibition [362]
AS	Atherogenic	D	↑ Atherosclerotic lesions [332]	↑ Glial cell activation [363]	↑ Memory impairments [363,364]
CAD	ApoE−/−	G	↑ Atherosclerotic lesions [342], cardiac hypertrophy [365]	↑ BBB permeability [366,367]	↑ Memory impairments [368]
CAD	Ldlr−/−	G	↓ Diastolic blood pressure, Cardiomyopathy [369] & ↑ Atherosclerotic lesions [370]	↑ BBB permeability [371], ↓ HPC cell proliferation & apoptosis [372,373]	↑ Memory impairments [373,374,375]

* Induction of systolic hypertension currently under debate in the scientific literature. ↑ and ↓ denote an increase or decrease respectively; G genetic intervention; D dietary intervention; TJP tight junction proteins; BBB blood–brain barrier; HPC hippocampus.

## Data Availability

Not applicable.

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
