# Peer review of "Investigating the Interplay between Cardiovascular and Neurodegenerative Disease"

_biology, 2024, doi:10.3390/biology13100764_

Round 1

Reviewer 1 Report

Comments and Suggestions for Authors

The manuscript provides a comprehensive overview of cardiovascular and related physiological changes observed in various neurodegenerative disease models, including Alzheimer’s Disease (AD), Parkinson’s Disease (PD), Multiple Sclerosis (MS), Amyotrophic Lateral Sclerosis (ALS), and Huntington’s Disease (HD). The discussion is well-organized and highlights significant findings from both rodent models and clinical observations. Here are some suggestions:

  1. The major strength of the manuscript is its comprehensive synthesis of studies across various disease models, offering a clear picture of cardiovascular dysfunction in neurodegenerative diseases. The detailed discussion includes relevant findings on autonomic dysfunction, blood-brain barrier integrity, and cell-autonomous disruptions in cardiomyocytes, effectively integrating these aspects to highlight their interconnected roles in disease pathology.
  2. While the manuscript acknowledges the potential for neurodegenerative diseases (NDDs) to impact cardiovascular diseases (CVDs), it currently lacks a thorough exploration of this bidirectional relationship. A more detailed discussion of how NDDs might influence CVDs, including potential mechanisms and shared risk factors, would provide a more comprehensive understanding of these complex interactions.
  3. In human disease section, ALS and its connection to CVDs is relatively brief and lacks depth. Expanding this section to include a more detailed analysis of current studies, potential mechanisms, and research gaps could significantly enhance the manuscript's completeness.
  4. Adding a general pictorial summary illustrating the relationships between various CVDs and NDDs, as well as shared mechanisms, could help readers better understand the complex interactions discussed in the manuscript.
  5. In animal section, the information regarding AD and its connection to CVDs is relatively sparse. Expanding this section to include a more detailed analysis of current studies, potential mechanisms, and research gaps would enhance the manuscript's completeness.
  6. In the animal studies section, the manuscript includes information about blood-brain barrier (BBB) integrity, RAS changes, and other factors but does not thoroughly explain how these aspects affect disease pathology. Adding a mechanistic analysis or a table summarizing these mechanisms could improve readers’ understanding.
  7. Both in the human and animal sections, the manuscript does not explain the mechanistic ways in which CVDs affect NDDs or vice versa. Including detailed mechanistic explanations would strengthen the discussion.
  8. The manuscript is well-designed, covering numerous human and animal studies, and the discussion section enhances the understanding of the findings. This section effectively integrates the reviewed literature and contributes to the overall impact of the manuscript.

Reviewer 2 Report

Comments and Suggestions for Authors

This is a well organized and written review. In some parts is a little repetitive but in general is useful and informative. I would suggest a list of abbreviations and to include some figure showing the potential common molecular mechanisms between degenerative diseases and cardiovascular dysfunctions.

Reviewer 3 Report

Comments and Suggestions for Authors

This is an extensive review of recent literature which makes a significant contribution to the field.  It is very well written.  There are some very minor 'typographical' issues such as Table 2, where "20-AUG-24 13:50:00" is inserted, and references 171 and 189 where authors are unclear.

This review explores the inter-relationship are cardiovascular disease and neurodegeneration. It initially reviews the clinical / epidemiological evidence and then considers animal models: behavioural changes in models of cardiovascular disease and cardiovascular changes in models of neurodegeneration.
The manuscript is novel in the extensive review of the interplay between the different disorders.
Other published material considers behavioural changes in specific models of cardiovascular disease, or cardiovascular changes in specific models of neurodegeneration. This is the first review to bring all of this information together.
The conclusions are sound: "Current research suggests that combinatorial models can enhance disease-specific phenotypes and may be a powerful tool to better model clinical disease."
The sources quoted are extensive and up-to-date.
The tables are useful summaries of the interactions highlighted within the text.
The manuscript is well-written in very good English. Some minor copy-editing is required within the list of references' in particular consistency of whether to give first names or just initials.
